# The Role of ncRNAs in the Immune Dysregulation of Preeclampsia

**DOI:** 10.3390/ijms242015215

**Published:** 2023-10-16

**Authors:** Carlos Mora-Palazuelos, Carlos Esteban Villegas-Mercado, Mariana Avendaño-Félix, Erik Lizárraga-Verdugo, José Geovanni Romero-Quintana, Jorge López-Gutiérrez, Saúl Beltrán-Ontiveros, Mercedes Bermúdez

**Affiliations:** 1Health Sciences Research and Teaching Center, Autonomous University of Sinaloa, Culiacan 80010, Sinaloa, Mexico; carlospalazuelos@uas.edu.mx (C.M.-P.); eriklizarraga@uas.edu.mx (E.L.-V.); saul.beltran@uas.edu.mx (S.B.-O.); 2Faculty of Dentistry, Autonomous University of Chihuahua, Chihuahua 31110, Chihuahua, Mexico; cmercado@uach.mx; 3Faculty of Dentistry, Autonomous University of Sinaloa, Culiacan 80010, Sinaloa, Mexico; marianaavendano@uas.edu.mx; 4Faculty of Chemical and Biological Sciences, Autonomous University of Sinaloa, Culiacan 80010, Sinaloa, Mexico; geovanniromero@uas.edu.mx; 5Faculty of Biology, Autonomous University of Sinaloa, Culiacan 80010, Sinaloa, Mexico; doctorjorgelopez@uas.edu.mx

**Keywords:** preeclampsia, ncRNA, miRNA, LncRNA, immune dysregulation, pregnancy

## Abstract

The main complications causing practically 75% of all maternal deaths are severe bleeding, infections, and high blood pressure during pregnancy (preeclampsia (PE) and eclampsia). The usefulness of ncRNAs as clinical biomarkers has been explored in an extensive range of human diseases including pregnancy-related diseases such as PE. Immunological dysregulation show that the Th1/17:Th2/Treg ratio is “central and causal” to PE. However, there is evidence of the involvement of placenta-expressed miRNAs and lncRNAs in the immunological regulation of crucial processes of placenta development and function during pregnancy. Abnormal expression of these molecules is related to immune physiopathological processes that occur in PE. Therefore, this work aims to describe the importance of miRNAs and lncRNAs in immune dysregulation in PE. Interestingly, multiple ncRNAS are involved in the immune dysregulation of PE participating in type 1 immune response regulation, immune microenvironment regulation in placenta promoting inflammatory factors, trophoblast cell invasion in women with Early-Onset PE (EOPE), placental development, and angiogenesis, promotion of population of M1 and M2, proliferation, invasion, and migration of placental trophoblast cells, and promotion of invasion and autophagy through vias such as PI3K/AKT/mTOR, VEGF/VEGFR1, and TLR9/STAT3.

## 1. Introduction

For optimal cellular function, the presence of a complex network of molecular factors is imperative. These factors intricately interact, ensuring precise control over gene expression. Recent advancements in molecular biology have elucidated that gene expression regulation is not solely governed by proteins but also by non-coding RNAs (ncRNAs) [1]. Due to the remarkable progress in the realm of next-generation sequencing and bioinformatics analysis, it has become feasible to identify a multitude of novel ncRNA molecules engaged in diverse physiological processes. Among these, microRNAs (miRNAs) and long non-coding RNAs (lncRNAs) play pivotal roles in transcriptional regulation across different levels [1].

MiRNAs are one of the most abundant and most studied natural single-stranded small ncRNAs of ~22 nucleotides that regulate gene expression [2]. In mammals, more than 60% of mRNAs have been hypothesized to be regulated by miRNAs, playing not only an important role in several cellular processes but also in the physiopathological processes of various diseases [3,4,5]. On the other hand, lncRNAs are another of the most widely studied ncRNAs, with lengths of more than 200 nucleotides, which play an important role in growth and development, and dysregulation has been associated with several diseases [3,6]. The usefulness of ncRNAs as clinical biomarkers is explored in an extensive range of human diseases such as cancer, diabetes, metabolic syndrome, immunological, neurological, infectious, and cardiovascular (CVDs) diseases [7], and have also been involved in pregnancy-related diseases such as PE [8,9].

In this regard, maternal mortality is unacceptably high; according to the World Health Organization, approximately 287,000 women died during pregnancy and childbirth in 2020, and every day die almost 800 women from preventable causes related to pregnancy and childbirth. The main complications causing practically 75% of all maternal deaths are severe bleeding, infections, and high blood pressure during pregnancy (PE and eclampsia), among others [10]. PE has been estimated in 2–8% of pregnancies globally [11] and produces new-onset hypertension (>140/90 mm Hg) mostly in the second half of pregnancy [12], accompanied with protein excretion in 24 h urine (>300 mg), or multiorgan maternal dysfunction, especially kidney and liver failure, neurological complications, thrombocytopenia, or hemolysis [13].

The pathogenesis of PE initiates with placental syncytiotrophoblast stress that causes excessive soluble fms-like tyrosine kinase-1 (sFlt-1) into the maternal circulation. Then, sFlt-1 binds to free placental growth factor (PlGF) or vascular endothelial growth factor (VEGF) (proangiogenic factors) with high affinity, thus preventing their interaction with their cell-surface receptors (i.e., the vascular endothelial growth factor receptor-1 (VEGR-1)) on the endothelial cells, leading to endothelial dysfunction and, subsequently, to systemic inflammation [14]. Traditionally, the diagnosis of PE relies on the detection of hypertension in two times (>140/90 mm Hg) and proteinuria (>300 mg/24 h). However, today, international organizations recommend diagnosis even in the absence of proteinuria when the pregnancy has evidence of multi-systemic damage [14]. PE is an enigmatic disorder of pregnancy that has been considered the “disease of theories”, and is known as the great obstetrical syndrome, in which several pathologic processes, and in some cases super-imposed, assist in the activation and recognition of the clinical symptomatology characteristics [14]. Evidence supports the involvement of placenta-expressed miRNAs and lncRNAs in the immunological regulation of crucial processes of placenta development and function during pregnancy, but their role remains unknown. However, it has been shown that abnormal expression of miRNAs and lncRNAs in the placenta is related to immune physio pathological processes that occur in PE. Therefore, this work aims to describe the importance of miRNAs and lncRNAs in immune dysregulation in PE.

## 2. Immune Dysregulation of Preeclampsia

The main cause of abnormal placentation in PE remains unclear, although genetic, immunological, and environmental factors have been associated [15]. A few theories support that immune response leading to PE may start as early as conception during contact with paternal-derived transplantation antigens during coitus and the consequent activation of regulatory T cells (Treg) [16]. In this regard, the fetal trophoblast acts as an alloantigen, producing a systemic inflammatory stage in the mother; however, in most cases, it is controlled. The interactions during the first trimester of gestation between decidual immune cells and trophoblast cells could be the cause of the inflammatory response onset; in the second and third trimester, a second inflammatory stage occurs and could be due to syncytiotrophoblast microparticles including pro-inflammatory cytokines, exosomes, anti-angiogenic agents, and cell-free fetal DNA that are released into the mother’s vascular system [12].

PE has two stages: first, poor trophoblastic invasion, a product of altered production of immunoregulatory cytokines and angiogenic factors; and second, a systemic, maternal-inflammatory response, where the endothelium is encouraged by necrotic and apoptotic syncytiotrophoblast cells into the maternal circulation [17]. This second stage could be associated with poor fetal growth and, consequently, in the development of intrauterine growth restriction [18].

The pathophysiology of PE is multifactorial, where the placental hypoxia is accompanied by high amounts of syncytiotrophoblast microparticles [17]. Furthermore, the possible activation of monocytes, dendritic cells (DCs), natural killer (NK) cells, and neutrophils by binding Toll-Like receptors (TLRs) with syncytiotrophoblast and damage-associated molecular pattern molecules during hypoxia enhances persistent inflammatory conditions [17]. B lymphocytes (B cells) can also participate in the pathophysiology by producing autoantibodies against adrenoreceptors and angiotensin-1 receptors. The risk of hypertension in PE is also related to neutrophil activation observed in endothelial dysfunction and extracellular trap, which consists of extracellular structures composed of chromatin and granular proteins released during the death process, which occurs upon neutrophil stimulation [17]. Moreover, the cytokines secreted by nonclassical, intermediate, and monocytes may generate inflammatory processes and changes in adaptative immune system cells. For all the above, it is believed that both innate and adaptive immune processes participate in the pathogenesis of PE, being responsible for poor placentation, an exacerbated inflammatory response, and endothelial dysfunction in Th1 immunity [19].

Even though PE development is affected by multiple immunologic factors, mechanisms, and pathways, one of the theories focuses on the study of immunological aspects as controls of immune responses in Th1, Th2, T helper 17 (Th17), and Treg. An abnormal immune response and high Th1/17:Th2/Treg ratio are “central and causal” to PE [20]. In this regard, is known that the balance in Th1/Th2 cytokines is important for maintaining the success of normal pregnancy [21]. In normal pregnancy, the production of the Th1 cytokine is inhibited and their overexpression predisposes one to PE development [21]. Previous reports support an excessive innate immune activity and a change toward a proinflammatory profile in PE [22]. For instance, preeclamptic patients show elevated Th1:Th2 ratios, involving increased secretion of proinflammatory cytokines, such as tumor necrosis factor alpha (TNF-α), interferon-gamma cytokine (IFN-γ), interleukin 6 (IL-6), and interleukin 17 (IL-17), and low interleukin 4 (IL-4) [23]. Furthermore, these placentas show suppression of interleukin 10 (IL-10), transforming growth factor beta receptor 1 (TGF-_β1_) and altered interleukin 2 (IL-2) IL-2/IL-10 and TNF-α/IL-10 ratios [24]. PE is also associated with an overexpression of interleukin 17 (IL-17) and interleukin 22 (IL-22), playing a critical role in the development of hypertensive syndrome. Previous studies have reported that decreased Tregs and increased Th17 CD4^+^ T cell subtypes may be involved in the pathophysiology of PE [20].

## 3. Non-Coding RNAs

NcRNAs do not play a direct role in gene coding and protein synthesis, but they assume the role of regulators for gene expression at epigenetic, transcriptional, posttranscriptional, translational, and posttranslational levels [3]. Biogenesis and functions of miRNAs and lncRNAs are shown in Figure 1.

**(a)** 
**miRNAs biogenesis and functions**


MicroRNAs are non-coding RNAs with a ~22 nucleotide length that are typically processed by RNA polymerase II/III in a post- or co-transcriptionally manner. A high proportion of identified miRNAs to date are intragenic and processed from intronic regions; nonetheless, there are a few which are transcribed from intergenic regions. In some cases, miRNA transcription results in clusters which share similar seed regions that are considered as miRNA families [25,26,27].

After processing by RNA polymerase, the first transcript consists of a hairpin structure (primary miRNA) which presents a base-paired stem structure; then, these are cleaved by a microprocessor containing Drosha (RNAase III-type) and its cofactor DGCR8, taking place in the nucleus [28]. Eventually, a hairpin structure of ~60–70 nucleotides in length, known as precursor miRNA (pre-miRNA), which contains a stem-loop structure is released and exported from the nucleus to cytoplasm through exportin-5 (EXP 5) and Ras-related nuclear protein guanosine triphosphate (RAN-GTP) [28]. Once pre-miRNA is in the cytoplasm, its terminal loop structure is cleaved by the action of Dicer and its cofactor TRBP, a ~20–22 nucleotide miRNA duplex containing two 5′ phosphorylated sequence strands, named a miRNA guide strand and its complementary passenger strand [28]. Then, the miRNA-duplex binds to the Argonaute protein to conform RISC (RNA-induced silencing complex), and the passenger strand is degraded [28]. Finally, the miRISC complex participates in gene expression regulation through the interaction of the miRNA seed region with the target messenger RNA 3′-UTR sequence, which leads to degradation or a blocking of mRNA to inhibit its translation [29].

One of the most important functions in which miRNAs participate is gene regulation, through mediating the degradation of mRNAs [30]. Transcription and translation are also meditated by miRNAs by two main mechanisms: the canonical pathway, which has been briefly mentioned above, and exerts degradation of mRNAs by the miRNA seed sequence [29]. Nonetheless, it has been reported that approximately 60% of mRNA-miRISC interactions depend on non-canonical pathways, supporting the idea that one miRNA sequence can target a plethora of mRNAs due to no complementary union among both sequences [31]. Thus, miRNA dysregulation can affect a vast number of functions in different biological processes.

Moreover, it has been shown that circulating microRNAs have pivotal roles in basic and clinical areas, and is known that fluctuations in circulating miRNAs have been associated with pathological processes such as inflammatory diseases [32,33], cancer [34], chronic diseases [35,36], as well as PE. Several studies have focused on the identification of biomarkers for the early detection of PE; for instance, plasma exosomal miR-517-5p, miR-520a-5p, and miR-525-5p (which belong to C19MC cluster pre-eclampsia) are down-regulated during the first trimester of gestation in women affected with gestational hypertension and PE, and were reported as a biomarker with high accuracy since their expression profile can identify women at risk of later development of gestational hypertension and PE by the first trimester [37]. Recently, there have been reported a battery of microRNAs that provided possible clinical applications since they were categorized according to PE severity [38]. In severe PE patients, miR-215, miR-155, miR-650, miR-210, and miR-21 were upregulated, and miR-18a and miR-19b1 present downregulation. In mild PE patients, miR-518b and miR-29a were found upregulated, while miR-15b and miR-144 were downregulated [38].

**(b)** 
**LncRNA biogenesis and functions**


It is quite important to know how lncRNAs are produced since their functional value and relevance are pivotal to maintaining homeostasis in biological processes. Unfortunately, their dysregulation is key to the development of illness. These are part of the ncRNAs, which are molecules of more than 200 nucleotides located in the nucleus and in the cell cytoplasm. They can interact with various proteins and other RNAs, inducing the direct or indirect regulation of gene transcription, as well as the translation of mRNA [39]. There is a whole genomic system involved in the biogenesis of lncRNAs, such as promoters, enhancers, and intergenic regions in eukaryotic genomes [40]. LncRNAs are classified into several types, regarding their functions as rRNA, tRNA, and cRNA. Regarding their subcellular localization, they can be nuclear lncRNAs, cytoplasmic lncRNAs, and mitochondrial lncRNAs [41] Cytoplasmic lncRNAs are low in abundance but very stable [42]; in the nucleus, they work to modulate transcription through interactions and chromatin remodeling [43,44], influencing the regulation of gene expression depending on their association with ribosomes, mitochondria, endoplasmic reticulum, the cell periphery, and phase separation bodies [45,46]. On the other hand, nuclear lncRNAs work in the nucleus to modulate transcription through interactions and chromatin remodeling [47]; these are abundant but very unstable [48], which is attributed to regulation by PABPN1, poly-A-polymerase-dependent hyperangilation, and the eventual breakdown of lncRNAs [46]. Typically, lncRNAs are transcribed by RNA polymerase II (Pol II), harboring a 5′ methyl-cytosine cap and 3′ poly-A-tail [49]. Hence, the biogenesis of lncRNAs is similar to that of mRNA since most of them suffer a process of capping, polyadenylation, as well as the typical splicing process [40]. On the other hand, they can be also processed by other non-canonical mechanisms, including cleavage by ribonuclease P (RNase P), resulting in matures 3′ ends. snoRNA-protein (snoRNP) performs the capping process, which leads to the generation of circular structures [40].

LncRNA biogenesis is performed by the conjunction of several regulators, various epigenetic modifications, and methylations of receptors. lncRNAs also regulate expression as decoys, acting as sponges or molecular sinks for transcription factors (TFs), RNA binding proteins, or microRNAs to enhance the activation of their target genes or by silencing them [50]. Guide lncRNAs are prone to bind to their target proteins and can lead to a ribonucleoprotein complex being guided to its specific target gene promoter or genomic loci; the regulation of gene expression at the transcription level occurs as a consequence [51]. Moreover, lncRNAs can act as scaffolds by assembling scaffolding complexes with TFs and effector molecules that can regulate the activity of RNA Pol II, histone modifications, as well as chromosome rearrangement; they also can regulate epigenetic and transcriptional control at gene expression levels [52]. Finally, the most recent mechanistic function of lncRNAs is SINEUPs, which are a class of antisense lncRNAs that enhance the translation of target mRNAs without affecting their levels; thus, it contains an inverted effector domain that confers biological activity and a binding site that provides target specificity [53].

Several studies have utilized a transcriptomic approach to pinpoint lncRNAs associated with PE. For instance, the expression level of lncRNA NORAD is related directly to hypertension in preeclampsia [54]. Nevertheless, the underlying molecular mechanisms by which NORAD is overexpressed and causes hypertension are still unknown. Another lncRNA, LOC391533, is related to sFlt-1 overexpression, giving the major phenotypes of preeclampsia, like hypertension and proteinuria [55]. He et al. employed microarray analysis and RNA profiling to identify738 differentially expressed lncRNAs out of 28,443 in PE placental tissues, compared to those from normal pregnancy [56]. They proposed that three specific lncRNAs (LOC391533, LOC284100, CEACAMP8) play a role in PE by regulating angiogenesis and vasculogenesis [56]. Using RNA-seq on decidual samples, Tong et al. identified 32 differentially expressed lncRNAs between normal and early-onset PE, 53 differentially expressed lncRNAs between normal and late-onset PE, and 32 differentially expressed lncRNAs between early-onset and late-onset PE, suggesting that distinct physiopathological mechanisms underlie early-onset and late-onset PE [57]. In a study focusing on the role of lncRNAs in early-onset PE development using microarray analysis, Wang et al. discovered 15,646 upregulated and 13,178 downregulated lncRNAs in the placenta of early-onset PE patients compared to preterm controls, revealing that the pathways overrepresented in early-onset PE patients were related to cell migration and cell motility [58].

## 4. miRNAs in the Immune Dysregulation of Preeclampsia

Recently, it has been described that miRNAs can be wrapped in exosomes, a subtype of extracellular small vesicles with varying sizes (20–130 nm) [59], to protect them from degradation by RNases [8]. These vesicles can be secreted to the systemic circulation from the placenta, resulting in multisystemic organ damage in patients with PE [8]. For the above, it is possible to detect circulating plasma exosomes which contain miRNAs as a diagnosis technique [60]. miRNAs involved in immune dysregulation in PE are shown in Table 1.

The human placenta-associated miRNAs are expressed in villous trophoblasts and secreted into maternal circulation via exosomes [61]. Several exosome-derived miRNAs have been associated with PE development such as miR-31-5p, which has been proposed as a potential biomarker to evaluate preeclampsia progression [61]. Additionally, exosomal miRNAs (Exo-miRNAs) miR-483-3p, hsa-miR-1237-3p, 365b-5p, hsa-miR-155-5p, hsa-miR-200b-3p, hsa-miR-342-3p, hsa-miR-140-3p, and hsa-miR-3909 can be found in umbilical serum, offering a pattern associated with microvascular dysfunction in mothers with PE [62]. Also, miR-210 expression in serum is increased during PE pregnancy progression, being associated with the severity of this disease [63] since its up-regulation is correlated with the inhibition of migration and the invasive capability of trophoblasts, and is linked to induction of the activity of several intracellular transcription factors [64].

Interestingly, miRNA analysis employing placental tissues showed a negative correlation between miR-126 (upregulated) and VCAM-1 (downregulated), a protein expressed by placental villous trophoblasts in the PE pregnancies group, proposing that this miRNA participates in the occurrence and development of EOPE through modulation of the invasion ability of trophoblast cells [65,66]. Another miRNA associated with the pathogenesis of PE is the upregulated miR-200b-3p, which contributes to the dysregulation of cell adhesion molecules (CAMs) and tight junction via profilin 2 (PFN2) regulation in placenta tissues [67]. Also, through in vitro analysis, it has been concluded that miR-146a-5p mediates trophoblast cell proliferation and invasion by Wnt2 expression regulation since this ligand promotes migration and proliferation of trophoblast cells via triggering the Wnt/β-catenin pathway [68]. In addition, the lncRNA DANCR activates the PI3K/AKT pathway by miR-214-5p downregulation, promoting the migration and invasion of chorionic trophoblast cells in PE [69].

In an immunological context, the cytokine TNFSF15, which has been identified in blood as a possible type 1 immune response during PE [70,71], can be regulated by miR-517a/b and miR-517c in extravillous trophoblast cells (EVTs) [72]. Studies have indicated that miR-146a regulates the immune microenvironment of the placenta by TGF-β/Smad4 pathway activation, promoting inflammatory factor expression in PE patients [73]. Additionally, let-7a expression in the placenta tissue of patients with severe preeclampsia (SPE) is significantly reduced, and the mRNA and protein levels of the important inflammatory factor TNF-α are significantly increased and, hence, significantly negatively correlated [74]. Interestingly, miR-145-5p regulates TNF-α expression by its upregulation in serum and mediating trophoblast cell invasion in women with EOPE [75].

Furthermore, a negative correlation between miR-203a-3p and IL-24 has been described in extract placental mononuclear cells and serum exosomes from PE patients, indicating that miR-203a-3p plays an important anti-inflammatory role in PE pregnant women [76]. The miR-548c-5p is downregulated and protein tyrosine phosphatase receptor type O (PTPRO) is upregulated in serum exosomes and placental mononuclear cells from PE patients, establishing a negative association [76]. Also, it has been reported that miR-548c-5p inhibits inflammation by IL-12 and TNF-α downregulation and less nuclear translocation of pNF-κB in macrophages [77]. Remarkably, serum IL-10 levels are decreased in women with PE, possibly contributing to systemic inflammation and decreasing the number of circulating CD4^+^CD25^+^CD127^−^ Tregs in women with PE. Also, placental tissues from PE patients showed a significantly decreased Foxp3 and significantly increased expression of miR-210, indicating a possibly positive co-regulation among them [78]. In vitro assays showed that under hypoxic conditions, human trophoblast cell-derived extracellular vesicles release miR-1273d, miR-4492, and miR-4417 to target HLA-G, mediating immune- and inflammation-related pathways and, consequently, triggering the development of PE [79].

**Table 1 ijms-24-15215-t001:** miRNAs involved in immune dysregulation in Preeclampsia.

Molecule	Target	Function	Reference
miR-517a/b and c	TNFSF15	Type 1 immune response regulation during PE	[69]
miR-146a	SMAD4	Immune microenvironment regulation in placenta promoting inflammatory factors expression in PE patient	[70]
let-7a	TNF-α	Participate in the occurrence and development of SPE	[71]
miR-145-5p	TNF-α	Mediates trophoblast cell invasion in women with EOPE	[72]
miR-203a-3p	IL-24	Anti-inflammatory role in PE pregnant women	[73]
miR-548c-5p	PTPRO	Anti-inflammatory factor in preeclampsia	[74]
miR-210	Foxp3	Association with maternal immune tolerance of the fetus by T-cells regulation	[75]
miR-1273d, miR-4492, and miR-4417	HLA-G	Mediate immune- and inflammation-related pathways promoting the development of preeclampsia.	[76]

## 5. LncRNAs in the Immune Dysregulation of Preeclampsia

LncRNAs have recently emerged as important regulators of gene expression since they can regulate, transcribe, and encode proteins, influencing the processes of molecules and cells in cancer biology, neoplasms, and inflammation, as well as in formation and development [80,81,82]. In particular, lncRNAs have emerged as molecules of great interest in PE, where they are involved in pathological pathways through the regulation of the expression of genetic and molecular factors, generating serious functional alterations in cell proliferation, differentiation, apoptosis, and migration in trophoblasts and cytotrophoblasts [9]. LncRNAs involved in immune dysregulation in PE are shown in Table 2.

Multiple reports have described that placental samples collected from women exhibit aberrant lncRNA expression in patients who are in labor or diagnosed with PE [80]. For instance, Yajuan Wang et al. examined placental tissues from pregnant women with and without preeclampsia, demonstrating that leptin gene LEP is involved in the JAK/STAT signaling pathway, placental development, and angiogenesis; when analyzing the infiltration of immune cells in patients with PE, it showed an increase in the population of M1 and M2 macrophages, compared to patients with normal pregnancy [80]. Furthermore, LEP is expressed in clinical samples of PE, as well as in the mouse model of PE and the HTR-8/SVneo cell line [83].

The lncRNA SH3PXD2A-AS1 is upregulated in term placentas from women with PE, influencing placental development by recruiting the CCCTC binding factor (CTCF) into promoter regions, thus inhibiting SH3PXD2A and C-C chemokine receptor type 7 (CCR7 transcription, which is a transcription factor involved in the invasion and migration of early trophoblast cells) [84]. Likewise, lncRNA INHBA-ASI inhibits trophoblast cell invasion and migration by restricting the transcription factor CENPB by limiting the binding of TNF receptor-associated factor 1 (TRAF1), contributing to the pathogenesis of PE, inducing lack of proliferation in trophoblasts in the placentation area [85].

LncRNA H19 is upregulated in placentas of PE patients in villous, cytotrophoblast, and interstitial trophoblast [86,87]. H19 overexpression in JEG-3 and HTR-8/SVneo cells reduces cell viability and promotes autophagy and invasion by activation of the PI3K/AKT/mTOR pathways [88]. Moreover, lncRNA SPRY4 (SPRY4-IT) regulates trophoblast cell migration without interfering with epithelium-mesenchymal tug, showing that it quickly joins Human Antigen R (HuR), which is an important RNA-binding protein in the cytoplasm that modulates gene expression by altering the stability of mRNA [89]. In addition, silencing of SPRY4-ITI in HTR-8/SVneo induces cell proliferation and migration by reducing the apoptotic response. Furthermore, in placentas with severe PE, the expression of SPRY4-ITI is higher than in normal placentas, suggesting that this lncRNA is associated with the pathogenesis of PE [89].

On the other hand, lncRNA HOTAIR modulates the progression of PE by inhibiting miR-106 in an EZH2-dependent manner [90]. Yan Zhang et al., identified a new regulatory mechanism by RNA, demonstrating a new pathway that governs the regulation of PUM1/HOTAIR in trophoblast invasion by downregulating the expression of lncRNA HOTAIR in the pathogenesis of PE [91]. Another lncRNA is LINC00922, which was highly expressed in PE tissues, significantly inhibiting the migration, proliferation, and invasion of HTR-8/SVneo cells, as well as the arrest of the cell cycle in the G_0_/G_1_ phase [92]. This affected the expression of cyclin-dependent kinase 2 (CDK2), G1/S-specific cyclin-D1 (cyclin D1), proliferating cell nuclear antigen (PCNA), matrix metalloproteinase 9 (MMP-9), vimentin, and E -cadherin [92].

There are also lncRNAs downregulated in the placenta, such as MALAT1, TUG1, MEG3, and HOXA11-AS, which show greater cell arrest and increased apoptosis, and a decrease in cell proliferation and migration [93,94,95]. HOXA11-AS silencing has been shown to regulate genes associated with trophoblast migration and proliferation through association with chromatin repressive factors such as Lsd1 and Ezh2 [96]. In the case of ZEB2-AS1 and TDGR1, they stimulate proliferation, while LINC00473 positively regulates invasion and migration by miR-15a-5p and Lsd1, all three in trophoblasts [97,98,99].

It has been shown that the silencing of MALAT1 in HTR-8/SVneo suppresses migration and invasion, but also, in coculture, inhibits tubule formation in endothelial cells due to the downregulation of the angiogenic factor VEGF [100]. MALAT1 directly influences maternal cells as it enhances the immunosuppressive properties of Mesenchymal Stem Cells (MSCs) in vivo, inducing M2 macrophage polarization mediated by indoleamine 2,3-dioxygenase (IDO) expression [101]. It is also known that overexpression of MALAT1 and other lncRNAs negatively influences the proliferation of dendritic cells (DC), regulatory T (Treg) cells, and Th1 cells [102,103]. Zhuang et al. mention that growth regulation by lncRNAs in DC influences the immune response and apoptosis mediated by TLR9/STAT3 signaling [104]. It is also known that MALAT1 regulates miR-206/IGF-1, inducing the invasion and migration of cytotrophoblasts via PI3K/Akt [105]. In addition, it was shown that it promotes trophoblast migration and invasion through the epithelial-mesenchymal transition induced via FOS, confirming that it promotes functions such as placental spiral artery remodeling [106]. These results are supported by Chen. et al., since they observed an evident decrease in the proliferation, invasion, and migration assays in JEG-3 cells [93]. In addition, a higher rate of apoptosis was observed due to cell cycle arrest in the G_0_/G_1_ phase, specifically, which was demonstrated with high levels of proteins such as caspases 3 and 9, cleaved polymerase-1 (PARP-1), and poly(ADP-ribose) [93].

**Table 2 ijms-24-15215-t002:** LncRNAs involved in immune dysregulation in Preeclampsia.

Molecule	Target	Function	Reference
LncRNA-miRNA-LEP	JAK/STAT signaling pathway in HTR-8/SVneo cell line.	Promotes placental development and angiogenesis, analyzing the infiltration of immune cells in patients with PE, promoting the population of M1 and M2, effectively inducing vascularization and immunomodulation of the inflammatory response compared to patients with normal pregnancy.	[80]
SH3PXD2A-AS1	CCCTC-binding factor (CTCF) into promoter regions, SH3PXD2A, CCR7.	Restricting the transcription factor CENPB by limiting the binding of TNF receptor-associated factor 1 (TRAF1), involved in the proliferation, invasion, and migration of placental trophoblast cells.	[81,82]
LncRNA H19	PI3K/AKT/mTOR pathways, Let-7	H19 regulates trophoblastic spheroid adhesion by competitively binding to let-7 and promotes invasion and autophagy via the PI3K/AKT/mTOR pathways in trophoblast cells.	[84,85]
SPRY4-IT1	Caspase-3, Bax and BCL-2	SPRY4-IT1 modulates proliferation, migration, network formation, and apoptosis, showing an increase in the expression of Caspase-3 and Bax and a reduction in the expression of Bcl-2 in trophoblast cells HTR-8/SVneo.	[86]
HOTAIR	miR-106, PUM1	HOTAIR modulates the progression of preeclampsia by inhibiting miR-106, an EZH2-dependent. In addition, upregulation of PUM1 affects trophoblast invasion by downregulating HOTAIR expression.	[87,88]
LINC00922	CDK2, cyclin D1, PCNA, MMP-9, vimentin, and E -cadherin.	Increased LINC00922 in preeclampsia regulates the proliferation, invasion, and migration of placental trophoblast cells, and arrest of the cell cycle in G_0_/G_1_ phase.	[89]
MALAT-1	VEGF/VEGFR1, IDO, TLR9/STAT3, regulates miR-206/IGF-1.	MALAT-1 is downregulated in preeclampsia and regulates JEG-3 trophoblast cell proliferation, apoptosis, migration, and invasion. It works through the VEGF/VEGFR1 signaling pathway and in mesenchymal cells it promotes immunosuppressive properties as well as proliferation and angiogenesis through the induction of VEGF and IDO. In Dendritic Cells, influences the immune response and apoptosis mediated by TLR9/STAT3 signaling. In addition, induced the invasion and migration of cytotrophoblasts via PI3K/Akt.	[90,97,98,101,102]
MEG-3	NF-κB, Caspase-3, and Bax	Inhibition of endogenous MEG-3 increases apoptosis and decreases migration of HTR-8/SVneo and JEG3 cells. In addition, MEG-3 influences the expression of NF-κB, Caspase-3, and Bax proteins in trophoblast cells. This could lead to aberrant conditions in HTR-8/SVneo and JEG3 trophoblastic cells, associated with uterine spiral artery remodeling and progression to preeclampsia.	[92]
HOXA11-AS	RND3 and HOXA7	HOXA11-AS regulate genes associated with trophoblast migration and proliferation through association with chromatin repressive factors such as Lsd1 and Ezh2.	[93]

## 6. Clinical Relevance

Proinflammatory cytokines have been studied from a clinical perspective to find a panel of biomarkers for diagnosing PE. In this context, proteins such as interleukin 8 (IL-8), C-reactive protein (CRP), TNFα, and IL-6 are possibly useful for identifying pregnant women with a probability of development of PE, mostly in the second and third trimesters [107]. However, there is not a single inflammatory biomarker for routine use to predict PE onset; although increased soluble fms-like tyrosine kinase-1 (sFlt-1) protein and decreased VEGF and placental growth factor (PlGF) proteins have been related to an early onset of PE, they are not used commonly as biomarkers in clinical practice. Therefore, a combination of biomarkers in conjunction with the identification of clinical risk factors is mandatory [108]. Consequently, a wide analysis of factors, such as ncRNA, that are affected in PE could help to predict PE.

Despite ncRNAs having become a revolutionary way to assess pathological process monitoring in several diseases, PE is yet to be a condition that warrants more investigation on ncRNA dysregulation, especially in immunological processes. Nonetheless, there is a plethora of ncRNAs that have been studied in tissue. Altogether, these findings might provide valuable data on how PE pathogenesis is regulated based on ncRNA expression [109]. Moreover, both lncRNAs and miRNAs have regulatory effects on the pathogenesis as lncRNAs influence the bioavailability of miRNAs in PE. An example of this behavior is lncRNA H19, which can promote miR-let7 while downregulating miRNA-675 expression, mechanistically promoting trophoblastic migration and invasion in vitro. It is also capable of diminishing TGF-β signaling [110,111]. On the other hand, cRNA MIR193BHG is upregulated in the serum of PE patients and might play a role in PE by competitively binding to SASH1 with miR-345-3p [112]. However, clinical research is still required to establish which ncRNAs could serve for the early detection of PE in pregnant women.

Clinically, another important way to find ncRNAs is through exosomes; these are abundant in blood, meaning that their analysis could be a diagnosis technique that is less invasive and accurate [113]. Also, these exosomes are secreted by trophoblasts at the implantation site and could have trophoblast-specific characteristics involved in cell proliferation and invasion. Specifically, these may contain molecules with characteristics of damaged trophoblasts, such as miRNAs [114], which provide a wide possibility as diagnostic or prognostic markers for several diseases; and, they could be useful for the prediction of PE development [115,116].

As has become evident, the invasion, migration, and proliferation of specialized cells in the placentation allow orderly physiological remodeling and angiogenesis; however, this process will be affected by deficient trophoblastic invasion, which influences poor decidualization and, consequently, aberrant artery remodeling spiral, leading to placental ischemia and severe PE [117,118]. LncRNAs are aberrantly overexpressed in PE placental tissues; this could provide a new target for its early diagnosis and treatment. In recent years, immunotherapy strategies have been developed because lncRNAs play a vital role in drug resistance [119] since they have great potential when evaluating regulatory mechanisms on tissues and cells, from DNA methylation [120], mechanisms of transcription factors and transcriptional regulation, as well as the regulation of their post-transduction targets [121,122]. However, there are limitations, such as specifically administering the respective molecules in the target cells, finding the precise lncRNA as a drug that is capable of not causing adverse effects on the expression of the regulatory complex, and, finally, elucidating the difference between the genes for encoding since lncRNAs are poorly conserved between species [123]. Therefore, lncRNA immunotherapy will depend on the subcellular microenvironment, the function of effector cells, and the decrease in protective cells, which would allow an effective strategy focused on PE.

## 7. Concluding Remarks

PE is yet to be a condition that needs more investigation on the ncRNA involved in immunological dysregulation. There is evidence showing that miR-517a/b and c, miR-146a, let-7a, miR-145-5p, miR-1273d, miR-4492, and miR-4417 participate in type 1 immune response regulation, immune microenvironment regulation in placenta promoting inflammatory factors, and trophoblast cell invasion in women with EOPE; meanwhile, miR-203a-3p, miR-548c-5p, and miR-210 have an anti-inflammatory effect or are associated with maternal immune tolerance of the fetus by T cell regulation.

Regarding lncRNAs, LncRNA-miRNA-LEP is involved in placental development and angiogenesis, promoting population of M1 and M2, effectively inducing vascularization; SH3PXD2A-AS1 is involved in the proliferation, invasion, and migration of placental trophoblast cells; H19 promotes invasion and autophagy via the PI3K/AKT/mTOR pathways in trophoblast cells; SPRY4-IT1 modulates proliferation, migration, network formation, and apoptosis; HOTAIR modulates the progression of preeclampsia through inhibiting miR-106, an EZH2-dependent; increased LINC00922 in preeclampsia regulates the proliferation, invasion, and migration of placental trophoblast cells; and HOXA11-AS regulate genes associated with trophoblast migration and proliferation through association with chromatin repressive factors such as Lsd1 and Ezh2. Meanwhile, downregulation of MALAT-1 through the VEGF/VEGFR1 signaling pathway in mesenchymal cells promotes immunosuppressive properties, as well as proliferation and angiogenesis through the induction of VEGF and IDO. In DC, MALAT-1 influences the immune response and apoptosis mediated by TLR9/STAT3 signaling. MEG-3 influences the expression of NF-κB, Caspase-3, and Bax proteins in trophoblast cells.

## Figures and Tables

**Figure 1 ijms-24-15215-f001:**
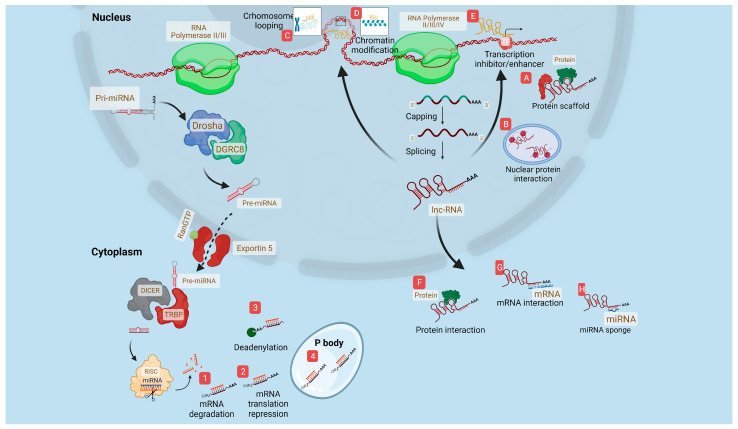
**Non-coding RNA biogenesis.** microRNA biogenesis and functions (1–4) exert mainly regulation of gene expression by interaction with mRNA. lncRNA biogenesis starts with RNA polymerase II/III/IV exerting several functions (A to H) like gene expression by sponging miRNAs, interactions, enhancers, or repressors of transcription factors, among others.

## Data Availability

Not applicable.

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
