# Peer review of "The Role of ncRNAs in the Immune Dysregulation of Preeclampsia"

_ijms, 2023, doi:10.3390/ijms242015215_

Round 1
Reviewer 1 Report
Comments and Suggestions for Authors
This manuscript shows the microRNAs and long non-coding RNAs involved in preeclampsia (PE), and many candidates proposed as useful markers in the pathological analyses and detection of signs. This paper will be valuable for the prediction and detection of PE. On the other hand, some inadequacies were found in the text and suggested corrections.
(Major comments)
1) How do the ncRNAs and lncRNAs play into the pathogenesis of hypertension and proteinuria? in PE? If possible, please add to the text.
2) EOPE is not well explained, please show the importance of EOPE in the pathogenesis or in the prevention and detection processes in PE.
3) Some terms were not well explained, and I could not understand their importance in PE. Please add some more explanation for the mechanism, physiological state, or importance in PE that these terms indicate. If they are not the main subjects in your manuscript, please consider deletion.
a) Line 64 "endothelial cell activation"
b) Line 65-66 "syncytiotrophoblast stress"
c) Line 67 "uteroplacental ischemia"
d) Line 114 “extracellular traps observed in endothelial dysfunction."
e) Line 356 "it quickly joins HuR" (use “HuR” in full spelling to clarify meaning)
4) Line 132: IL-6 belongs to Th2 cytokine, so there is a contradiction in the description (describing the induction of Th1 dominant condition)
5) Line 383: It is not indicated what MSC is an abbreviation for. Furthermore, If MSCs mean “mesenchymal stem cells”, then it feels wrong that MSCs have immunosuppressive properties.
(Minor comments)
1) The following terms are of uncertain meaning. Full spelling is needed.
a) Line 286: CAMs.
b) Line 286: PFN2
c) Line 307: PTRPO
d) Line 344: CCR7
2) There are corrections in the following words and sentence.
a) Line 304: IL24 --> IL-24
b) Table 1 (miR-210): T cells regulation --> T-cell regulation
c) Table 2 (LncRNA-miRNA-LEP): while --> While
d) Table 2 (SPRY4-IT1): HTR-8SV/neo --> HTR-8/SVneo
e) Table 2 (Meg-3): The sentence is not complete in "inhibition of endogenous MEG-3 increased apoptosis and decreased migration of HTR-8/SVneo and JEG3 cells”.
f) Lnc --> lnc: Line 326, 364, 462, 464, 465, 477
3) Line 478-486: The relationship between subjects and predicates is complicated in those descriptions. Please indicate what the important effects or targets lncRNAs have in the pathogenesis of PE. For those effects and targets, please indicate what specific lncRNAs are involved and describe the important ones.
Comments on the Quality of English Language
The subject-predicate relationships in lines 477-486 are difficult to understand.
Author Response
We acknowledge to editors and reviewers for their kind observations regarding our manuscript. All your recommendations have been followed and answered. Changes in the manuscript can be reviewed in the track changes of word. Please see the word file attached.

Reviewer 2 Report
Comments and Suggestions for Authors
This paper specifically connects ncRNA regulation with immune system dysfunction in preeclampsia, which is a less-explored angle. The article needs some feedback regarding methodological improvements and controls that the authors could consider:
- Many cited works lack detail on gestational age and comorbidities of PE patients. Highlighting the need for stratification by these variables would improve reproducibility and interpretation.
- Use well matched healthy pregnant controls (same gestational age, BMI, age, parity) to reduce confounding.
- Instead of single time-point sampling, follow patients throughout pregnancy to distinguish causal ncRNA changes from consequences of established PE.
- Validate ncRNA findings in diverse cohorts (different ethnic backgrounds, geographical regions) to account for population heterogeneity.
In summary, while the study is methodologically solid with its prospective, controlled, and randomized design, integrating these improvements would significantly enhance the internal and external validity, reduce bias, and provide stronger evidence for clinical recommendations.
Author Response
Dear Editor and Reviewer,
Thank you sincerely for your valuable time and thoughtful comments on our manuscript titled “The Role of ncRNAs in the Immune Dysregulation of Preeclampsia.” We appreciate the methodological suggestions, which highlight essential concerns regarding rigor and reproducibility in primary studies on ncRNAs in preeclampsia.
However, we would like to respectfully clarify a scope point. This work is a narrative review designed to synthesize and discuss existing evidence on non-coding RNA (ncRNA) regulation and immune dysfunction in preeclampsia. By its nature, a narrative review does not present new experimental data or original patient cohorts that could be stratified, re-sampled, or revalidated at this stage. While recommendations such as stratifying patients by gestational age and comorbidities, using well-matched controls, implementing longitudinal sampling, or validating findings across different populations are highly relevant, they relate to the design of future primary research, not corrections applicable to our manuscript.
Thank you again for your constructive feedback. We remain available for any further clarification.